mental health; community-based initiatives; digital mental health; Covid; group interventions

**Corresponding author:**
Leah E. James;
Email: leahemilyjames@gmail.com

# Feasibility, acceptability and preliminary effectiveness of a community-based group psychosocial support model for conflict survivors in Colombia: An assessment of in-person and remote intervention modalities during the COVID-19 pandemic

Leah E. James[1,2] ⬛, Nicolás García Mejía[3,4] ⬛, Juan F. Botero-García[5] and Michel Rattner[3,6] ⬛

[1]Heartland Alliance International, Chicago, IL, USA; [2]Institute of Behavioral Science, University of Colorado, Boulder, CO, USA; [3]Department of Psychology, Universidad de Los Andes, Bogota, Colombia; [4]Department of Clinical Psychology and Experimental Psychopathology, Faculty of Social and Behavioral Sciences, University of Groningen, Groningen, Netherlands; [5]Heartland Alliance International, Cali, Colombia and [6]Department of Psychology, Palo Alto University, Palo Alto, CA, USA

## Abstract

**Background:** Community-based psychosocial support (CB-PSS) interventions utilizing task sharing and varied (in-person, remote) modalities are essential strategies to meet mental health needs, including during the COVID-19 pandemic. However, knowledge gaps remain regarding feasibility and effectiveness.

**Methods:** This study assesses feasibility, acceptability and preliminary effectiveness of a CB-PSS intervention for conflict-affected adults in Colombia through parallel randomized controlled trials, one delivered in-person ($n = 165$) and the other remotely ($n = 103$), implemented during the COVID-19 pandemic and national protests. Interventions were facilitated by nonspecialist community members and consisted of eight problem-solving and expressive group sessions.

**Findings:** Attendance was moderate and fidelity was high in both modalities. Participants in both modalities reported high levels of satisfaction, with in-person participants reporting increased comfort expressing emotions and more positive experiences with research protocols. Symptoms of depression, anxiety and posttraumatic stress disorder improved among in-person participants, but there were no significant changes for remote participants in comparison to waitlist controls.

**Implications:** This CB-PSS intervention appears feasible and acceptable in both in-person and remote modalities and associated with reduction in some forms of distress when conducted in-person but not when conducted remotely. Methodological limitations and potential explanations and areas for future research are discussed, drawing from related studies.

## Impact statement

Although a significant body of research supports the use of community-based mental health and psychosocial support interventions for conflict survivors, gaps remain regarding evidence for group models utilizing task-shifting and varied (in-person, remote) modalities. This study assesses the feasibility, acceptability and preliminary effectiveness of a community-based psychosocial support group intervention for conflict-affected adults in Colombia through parallel trials, one delivered in-person and the other remotely, implemented during the COVID-19 pandemic and national protests. Interventions were facilitated by trained, nonspecialist community members and consisted of eight weekly problem-solving and expressive group sessions of approximately 120 min each. Attendance was moderate and fidelity was high in both the modalities. Participants in both modalities reported high levels of satisfaction with the intervention, but those in the in-person group reported higher levels of comfort with emotional expression and positivity about some research procedures. In these samples, in-person group participation was effective in reducing the symptoms of depression, anxiety and posttraumatic stress disorder. However, no effects were found for remote group participation in comparison to control. Results suggest that trained community members can meaningfully impact the mental health of their peers when interventions are implemented in-person but raise important questions about the use of remote modalities for psychosocial support groups. When examined alongside qualitative data collected through companion studies, the results suggest that

additional work is needed to identify best practices to ensure that remote group interventions engage participants, protect confidentiality and facilitate emotional expression and exchange of peer support. Methodological limitations are discussed, including the fact that participants were able to choose whether to participate in-person or remotely; future work with randomization to modality is recommended. This work offers key insights for informing future research and optimal scale-up of this and related community-based psychosocial support models in Colombia and globally.

## Introduction

A growing body of evidence demonstrates an increased risk of mental health and psychosocial problems among populations exposed to conflict and displacement (Mesa-Vieira et al., 2022; Carpiniello, 2023). Coexisting stressors associated with life in many low- and middle-income countries (LMICs), such as poverty, unemployment, limited access to education and sociopolitical instability can exacerbate risk (Rathod et al., 2017; Alloh et al., 2018). The COVID-19 pandemic has further intensified negative mental health outcomes globally (Kola et al., 2021; Wu et al., 2021). Despite a clear need, LMICs often suffer from insufficient human and financial resources to provide mental health and psychosocial support (MHPSS) services; some estimates suggest that close to 75% of those requiring care are unable to receive it (World Health Organization [WHO], 2021).

Colombia has struggled with an internal armed conflict persisting over five decades, conducted amidst widespread poverty and inequality (Cuartas Ricaurte et al., 2019). Despite the signing of a peace agreement in 2016, violence against civilians continues due to disputes between dissident illegal armed forces, narcoterrorism and insufficient enforcement of the peace agreement (Barragan, 2017; Nilsson and González Marín, 2020). As of May 2023, the Colombian Victims Unit Registry reported more than 12 million violent events and over 9 million victims, the majority of whom had been internally displaced (Unidad para las Víctimas (UV), 2024). Moreover, refugees fleeing Venezuela's civil conflict and associated economic troubles have flooded into Colombia; as of 2023, 2.4 million Venezuelan refugees are registered in Colombia's temporary protection statute (Migración Colombia, 2023).

Research has shown high levels of psychological distress, including symptoms associated with posttraumatic stress disorder (PTSD), generalized anxiety and depression as well as impaired functioning in Colombian internally displaced populations and other armed conflict victims (Bell et al., 2012; Campo-Arias et al., 2014; Gómez-Restrepo et al., 2016; Cuartas Ricaurte et al., 2019; Castro-Camacho et al., 2023). Colombia's 2016 peace agreement includes provisions for implementation of MHPSS services for victims and perpetrators of violence, emphasizing the need for culturally adapted intervention approaches, coordination between national, regional and local agencies, capacity strengthening of MHPSS service providers, including nonspecialist/lay providers and integration of global initiatives such as the Mental Health Gap Action Program (WHO, 2016a; Idrobo et al., 2018). Ensuring equitable access to MHPSS services among victims is considered critical for the success of the peace process but enforcement is highly challenging (Idrobo et al., 2018). The COVID-19 pandemic has further impeded progress by severely impacting access to public services, as well as both formal and informal employment opportunities and the financial stability of the community overall (Gillies et al., 2021).

Increasingly, community-based MHPSS interventions have been employed to promote well-being and to prevent the development and exacerbation of symptoms associated with mental health conditions in LMICs, with encouraging effects (for a review, Barbui

et al., 2020). In Colombia, promising interventions include psychosocial support group models based in community settings, which draw on community strengths and practices to collectively address problems and facilitate exchange of peer support (Pacichana-Quinayáz et al., 2016; Osorio-Cuellar et al., 2017; Aranguren-Romero and Rubio-Castro, 2018). In some cases, community-based MHPSS activities adopt a task-sharing approach, in which facilitation of activities is conducted by trained nonspecialist community members in an effort to fill gaps resulting from the limited number of professionals in many LMIC community settings (WHO, 2008). This approach has the potential to facilitate cultural adaptation, enhance community buy-in, help to build community capacity and support sustainability (Javadi et al., 2017; Le et al., 2022). A growing global evidence base demonstrates the feasibility, acceptability and effectiveness of MHPSS interventions provided by nonspecialists in community settings, especially when facilitators have strong community ties and are supported through comprehensive training and consistent supervision (Raviola et al., 2019; Le et al., 2022). However, task-sharing approaches also entail challenges, and limited evidence exists regarding the effectiveness of such models, including in Colombia and the Latin American region and when services are provided remotely (e.g., Le et al., 2022).

In the context of the COVID-19 pandemic, additional innovative methods have been employed to increase access to services (Moreno et al., 2020; Armijos et al., 2023). Remote service delivery through tele-mental health and/or digital tools is a key strategy to enhance the uptake of services during circumstances that impede in-person participation, including pandemics, conditions of community violence and when travel is otherwise difficult (e.g., in some rural areas) (Fu et al., 2020; IFRC, 2020) and was used widely during the COVID-19 pandemic (Witteveen et al., 2022). Such approaches also have the potential to reduce exposure to stigma and attenuate barriers to intervention attendance caused by work and family commitments, which can be especially problematic in impoverished communities (Sijbrandij et al., 2017; Naslund et al., 2019). However, remote approaches also introduce significant challenges and little work has examined feasibility or effectiveness of community support group models utilizing remote modalities (Ibragimov et al., 2022).

The current study aims to contribute to filling these gaps by assessing a community-based group psychosocial support (CB-PSS) intervention utilizing a task sharing approach delivered through two parallel trials, one conducted in-person and one remotely, to conflict survivors residing in Colombia's Pacific Coast. This intervention, facilitated by Community Psychosocial Agents (CPAs) (nonprofessional members of the community with prior training and experience providing PSS), aims to reduce distress and functional impairment and enhance community resilience through use of collective problem solving and emotional regulation activities. The current study examines the feasibility, acceptability and preliminary effectiveness of both the in-person and remote trials. Additionally, it explores demographic and baseline mental health predictors of attendance and moderators of outcomes to better understand when and for whom this intervention is likely to be

feasible and effective and compares findings of in-person and remote modalities.

## Methods

### Study design

This study utilizes a randomized controlled trial design (registration ICRTSN32986363) to test a community-based psychosocial support group (CB-PSS) intervention ("Grupo de Apoyo Comunitario") provided in two delivery modalities: in-person and remote. For ethical and accessibility reasons, participants were able to choose whether to participate in the in-person group trial or in the remote group trial. The study was originally planned as a single RCT, but in light of participant choice of modality, results for each modality are presented as separate trials for ease of interpretation. Participants in the waitlist control condition were offered the interventions after completing the second assessment interview (approximately five months after enrollment).

### Setting

This research was conducted in Quibdó, the capital city of the Chocó department and the rural community of Tutunendo (about 15 min northeast of Quibdó), Colombia. This region, located on the Colombian Pacific coast, has long been affected by the country's internal armed conflict and drug trafficking, which have contributed to widespread corruption and poverty. Nearly 65% of Choco residents live in poverty compared to 39% nationally (Departamento Nacional de Planeación, República de Colombia, 2023). According to Colombia's National Victims Registry (UV, 2024), more than 529,000 individuals in Chocó were registered as victims of the country's armed conflict as May 2023; of these, 26% are direct victims of forced displacement and 24% report witnessing conflict-related homicide. Additionally, as of May 2023, more than 3,900 Venezuelans reside in Chocó and are registered under Colombia's temporary protection statute (Migración Colombia, 2023). Colombia suffers from an inequitable distribution of health services (including MHPSS), with Chocó being one of the departments with the greatest shortage (Rojas-Bernal et al., 2018; WHO, 2021).

During the experimental phase of this study (March–August 2021), rates of COVID-19 infection were high in Quibdó, peaking in May 2021 (Instituto Nacional de Salud, 2023). The National Health Institute calculated that the incidence rate for the municipality was 10,257 for 100,000 inhabitants, the highest in the department. In addition, between April and June 2021, Colombia experienced country-wide protests against the government and associated police violence and human rights violations, resulting in food and gas shortages, impeding transportation and potentially affecting well-being more generally (Naciones Unidas, 2021).

### Participants

Participants were recruited by CPAs using a non-probabilistic snowball sampling approach, drawing from their networks of local organizations including neighborhood associations, migrant associations and associations of victims of the armed conflict, as well as through the Women's Department at the Quibdó Mayor's office (a body dedicated to promoting essential services for women). In some cases, CPAs presented the research opportunity to community members during meetings of these organizations, and in others, organizational leaders provided CPAs with contact lists of

community members expressing interest (with participant permission). CPAs asked potential participants to share information about the opportunity with others who might be interested.

Adult (age 18 or over) residents of Quibdó and the rural community of Tutunendo who reported that they were exposed to conflict violence (assessed by self-report during the recruitment process) were eligible to participate. Community members who had participated in Association of Organizations for Emotional Support's (ACOPLE's) MHPSS services in the past were excluded, as were those reporting significant risk of suicide/self-harm (measured through Heartland Alliance International's [HAI] suicide risk assessment) or potential psychosis (determined by interviewer perception of hallucination, delusion or disordered thought in the participant), both assessed during the pre-intervention interview. Referrals to the National Health Service were made for individuals requiring further care. CPAs used a written recruitment script to share the opportunity with potential participants. Those who expressed interest completed an informed consent process (see Ethics section) and provided verbal consent. CPAs then asked participants to choose their preferred intervention modality (in-person or remote) and scheduled pre-intervention interviews.

The decision to give participants the opportunity to choose their modality was made based on the results of a pilot study (Rattner et al., 2023) in which the primary lesson learned was that participants wished to decide for themselves whether to participate in MHPSS services in-person or remotely. Participants and staff shared that determining whether to participate in-person or remotely is a highly personal decision with important implications for accessibility, particularly in an emergency context (Armijos et al., 2023). The research team determined that randomly assigning participants to modality would compromise equitable access to services and therefore that allowing participants to make this choice was a more ethical option amid the COVID-19 and national protest emergencies. The sample size was calculated to provide sufficient participant numbers in each trial (in-person and remote) to produce medium effect sizes, while allowing for the potential that different numbers of participants would choose to join each modality and for participant attrition. A total of 165 in-person and 103 remote participants consented to participate in the study and completed pre-intervention assessment interviews (see Table 1).

### Randomization

Participants were based in 27 neighborhoods of Quibdó within six *comuna* (an administrative division used to group neighborhoods in Colombia). Participants within each comuna were individually randomized to intervention and waitlist control conditions. This approach allowed for feasible transport and aligned with the 'community-based' focus of bringing together members of the same community. Randomization was done by the research manager using EXCEL's RAND function. In-person modality randomization resulted in 82 intervention participants (distributed into 10 intervention groups) and 83 waitlist control participants. Remote modality randomization resulted in 52 intervention participants (distributed into eight intervention groups) and 51 waitlist control participants. Waitlist control participants received the same intervention between August and November 2021 (see Figure 1).

### Blinding

To allow for blinding to condition among interviewers, assessment interviews were conducted by different CPAs than those who facilitated the intervention groups for those participants. It was

**Table 1.** In-person and remote intervention participant demographics

| | In-person | | | Remote | | |
|---|---|---|---|---|---|---|
| | Control | Experimental | Overall | Control | Experimental | Overall |
| | (*n* = 83) | (*n* = 82) | (*n* = 165) | (*n* = 51) | (*n* = 52) | (*n* = 103) |
| **Demographics** | | | | | | |
| **Age** | | | | | | |
| Mean (SD) | 39.6 (14.0) | 42.1 (17.7) | 40.8 (16.0) | 35.4 (11.4) | 35.3 (12.3) | 35.4 (11.8) |
| **Gender** | | | | | | |
| Men | 12 (14.5%) | 8 (9.8%) | 20 (12.1%) | 5 (9.8%) | 6 (11.5%) | 11 (10.7%) |
| Women | 71 (85.5%) | 74 (90.2%) | 145 (87.9%) | 46 (90.2%) | 46 (88.5%) | 92 (89.3%) |
| **Area of residence** | | | | | | |
| Rural | 29 (34.9%) | 24 (29.3%) | 53 (32.1%) | 2 (3.9%) | 4 (7.7%) | 6 (5.8%) |
| Urban | 54 (65.1%) | 58 (70.7%) | 112 (67.9%) | 49 (96.1%) | 48 (92.3%) | 97 (94.2%) |
| **Nationality** | | | | | | |
| Colombian | 72 (86.7%) | 71 (86.6%) | 143 (86.7%) | 38 (74.5%) | 43 (82.7%) | 81 (78.6%) |
| Venezuelan | 11 (13.3%) | 11 (13.4%) | 22 (13.3%) | 13 (25.5%) | 9 (17.3%) | 22 (21.4%) |
| **Education** | | | | | | |
| Undergraduate degree or higher | 20 (24.1%) | 24 (29.3%) | 44 (26.7%) | 19 (37.3%) | 23 (44.2%) | 42 (40.8%) |
| Primary school or less | 17 (20.5%) | 23 (28.0%) | 40 (24.2%) | 10 (19.6%) | 13 (25.0%) | 23 (22.3%) |
| Middle to high school | 45 (54.2%) | 35 (42.7%) | 80 (48.5%) | 22 (43.1%) | 16 (30.8%) | 38 (36.9%) |
| Missing | 1 (1.2%) | 0 (0%) | 1 (0.6%) | na | na | na |
| **Marital status** | | | | | | |
| Single | 27 (32.5%) | 24 (29.3%) | 51 (30.9%) | 18 (35.3%) | 21 (40.4%) | 39 (37.9%) |
| Married or partnered | 53 (63.9%) | 50 (61.0%) | 103 (62.4%) | 31 (60.8%) | 28 (53.8%) | 59 (57.3%) |
| Divorced, separated or widowed | 3 (3.6%) | 8 (9.8%) | 11 (6.7%) | 2 (3.9%) | 3 (5.8%) | 5 (4.9%) |
| **Ethnicity** | | | | | | |
| Afro–descendant | 72 (86.7%) | 74 (90.2%) | 146 (88.5%) | 38 (74.5%) | 43 (82.7%) | 81 (78.6%) |
| Indigenous | 11 (13.3%) | 8 (9.8%) | 19 (11.5%) | 9 (17.6%) | 8 (15.4%) | 17 (16.5%) |
| Missing | na | na | na | 4 (7.8%) | 1 (1.9%) | 5 (4.9%) |
| **Work status** | | | | | | |
| Informal (no contract) | 36 (43.4%) | 28 (34.1%) | 64 (38.8%) | 21 (41.2%) | 12 (23.1%) | 33 (32.0%) |
| Formal (contracted) | 5 (6.0%) | 4 (4.9%) | 9 (5.5%) | 5 (9.8%) | 9 (17.3%) | 14 (13.6%) |
| Work at home (domestic duties, childcare) | 20 (24.1%) | 22 (26.8%) | 42 (25.5%) | 13 (25.5%) | 21 (40.4%) | 34 (33.0%) |
| Unemployed | 20 (24.1%) | 23 (28.0%) | 43 (26.1%) | 7 (13.7%) | 3 (5.8%) | 10 (9.7%) |
| Student | 2 (2.4%) | 5 (6.1%) | 7 (4.2%) | 5 (9.8%) | 7 (13.5%) | 12 (11.7%) |
| **Displaced** | | | | | | |
| Yes | 54 (65.1%) | 57 (69.5%) | 111 (67.3%) | 33 (64.7%) | 41 (78.8%) | 74 (71.8%) |
| | In-person | | | Remote | | |
| | Control | Experimental | Overall | Control | Experimental | Overall |
| **Baseline mental health outcome variables** | | | | | | |
| Depression | 0.98 (0.55) | 1.18 (0.60) | 1.06 (0.58) | 1.17 (0.59) | 1.12 (0.63) | 1.15 (0.61) |
| Anxiety | 0.83 (0.62) | 1.02 (0.71) | 0.90 (0.66) | 0.99 (0.67) | 0.96 (0.73) | 0.95 (0.69) |
| PTSD | 1.05 (0.67) | 1.15 (0.64) | 1.09 (0.65) | 1.07 (0.67) | 1.07 (0.71) | 1.05 (0.67) |
| Generalized distress | 2.62 (0.86) | 2.65 (0.85) | 2.64 (0.87) | 2.67 (0.81) | 2.77 (0.74) | 2.73 (0.79) |
| Functional impairment | 1.48 (0.42) | 1.55 (0.46) | 1.51 (0.44) | 1.65 (0.59) | 1.68 (0.65) | 1.66 (0.61) |
| Community resilience | 3.49 (0.39) | 3.44 (0.45) | 3.46 (0.43) | 3.50 (0.42) | 3.42 (0.55) | 3.44 (0.50) |

not possible to blind participants to condition due to the nature of the intervention.

## Intervention

The CB-PSS model assessed in this study is based on an intervention approach originally developed through the ACOPLE program funded by the United States Agency for International Development (USAID). HAI implemented ACOPLE from 2010 to 2020 in partnership with the National Association of Displaced Afro-Colombians (AFRODES), the Institute for Research and Development in the Prevention of Violence and Promotion of Social Coexistence (CISALVA) based at the Universidad del Valle, and Johns Hopkins University. ACOPLE provided individual (Bonilla-Escobar et al., 2018) and group (Osorio-Cuellar et al., 2017) MHPSS services delivered by CPAs with training and supervision by professionals to conflict-affected communities of Afro-Colombian descent on Colombia's Pacific Coast.

Whereas ACOPLE's interventions were initially focused on resolving trauma-related reactions associated with the armed conflict, in later years the group model evolved to increase focus on community problem-solving and culturally informed expressive activities to accommodate needs shared by participants and staff. As part of the current project, an updated intervention protocol was drafted by HAI's Colombian MHPSS professionals in collaboration with CPAs, and subsequently, an in-depth facilitation guidance with detailed examples of how each session could be run was developed with CPA leadership (HAI, 2020, 2021). The current intervention ("Grupo de Apoyo Comunitario") consists of eight sessions, including an introductory session and three collaborative problem-solving sessions interspersed with four expressive sessions drawing from culturally informed artwork and dance, designed to strengthen emotional regulation. Problem-solving sessions were informed by WHO's (2016b) Problem Management Plus protocol and consisted of collaboratively listing problems shared by group members, choosing and defining a problem for discussion, considering ideas nonjudgmentally and selecting useful strategies, developing an action plan and finally reviewing outcomes in a subsequent session. Expressive sessions entailed the identification and sharing of emotions through creative activities such as drawing a mandala, creating a paper mask, dance and body movement and creating a 'heroes' story. Each session included a relaxation or visualization activity (e.g., butterfly hug or safe space visualization) and frequent opportunities for exchange of peer support. While activities were the same in both the in-person and remote modalities, the manuals also included guidance for adaptation to a remote format, for example, ways to incorporate the "chat" function on Zoom to enhance peer interaction and tips for supporting participants to complete artwork independently using materials delivered to their homes and then to share with peers using their phone cameras.

Each intervention group was facilitated by two CPAs (eight total CPAs). All CPAs were women of Afro-Colombian descent and members of the Quibdó community known for their work in neighborhood- or municipality-level women's and victims' organizations. All had prior training in the ACOPLE project and experience facilitating ACOPLE MHPSS activities; some had originally been participants in these activities before applying to facilitate. CPAs participated in two weeks of intensive training (one prior to the pilot and another before the RCT) on the updated CB-PSS group protocol, and weekly group supervision conducted by MHPSS professionals. They were accompanied in sessions by either a psychologist or a social worker who monitored sessions and provided feedback for discussion in supervision sessions. Session monitors also completed fidelity checklists to assess whether all CPAs completed all components planned for that session.

Ten in-person intervention groups and eight remote groups were conducted, with seven to ten participants per group. In-person groups were conducted in centrally located community centers with adherence to biosecurity protocols while remote groups were conducted through online calls using the Zoom platform. Remote participants received mobile internet credit prior to each session (approximate value $2) and were able to borrow smartphones if needed (25 participants did so). Before the groups began, CPAs completed remote service safety planning checklists with participants, designed to encourage private and safe participation (e.g., to prevent being overheard by household members or coworkers) and provided brief training on use of Zoom. In-person participants received travel funds sufficient for nonpublic transport to reduce risk of COVID-19 exposure (approximate value $1) as well as face masks and hand sanitizer for use during group sessions. Both groups received snacks and supplies for each session (approximate value $2.5); these were delivered to the homes of remote participants.

Some staff and participants were infected by COVID-19 during the implementation period, resulting in delays in session scheduling both for the in-person and remote groups. In-person groups were more affected, pausing for approximately three weeks on average, with one group suspended for eight weeks. No changes were made to the intervention protocol.

## Feasibility

Feasibility was assessed through 1) participant attendance, which was documented by CPAs and observers during each group session, and 2) intervention fidelity, using fidelity checklists completed by psychologists or social workers observing each group session. Fidelity checklists were developed by the research team based on the intervention protocol and consisted of between 15 and 21 key activities for each of the eight sessions, each of which was checked off as completed or not completed by observers (i.e., supervisors) in each group session. A total fidelity score representing the percentage of sessions attended was calculated for each intervention group.

## Acceptability and effectiveness outcomes

All participants were interviewed prior to the intervention (PRE, March–April 2021) and after the intervention group had finished the intervention (POST, July–August 2021). The PRE assessment included three sections: 1) sociodemographic measures; 2) primary and secondary outcome measures (see Table 2) and 3) risk screening measures (suicide and self-harm, exposure to and perpetration of violence and psychosis, used to confirm eligibility). The POST assessment included the primary and secondary outcome measures, as well as the Reactions to Research Participation Questionnaire (RRPQ) (Newman et al., 2001). Intervention participants also completed an intervention acceptability survey of 13 items. The RRPQ and the acceptability survey were used to assess acceptability of both research and intervention activities, while the outcome measures were used for preliminary assessment of intervention effects. Primary outcome measures assessing distress (anxiety, depression and PTSD) have been used historically with the ACOPLE project, while additional measures were added as part of a

**Table 2.** Mental health outcome measures

| Variable | Scale | Instrument | Description | Cronbach α |
|---|---|---|---|---|
| *Primary outcome measures* | | | | |
| Depression | 0–3 | Hopkins Symptoms Checklist –HSCL–25 (Derogatis et al., 1974) | 25 items, 15 for depression symptoms and 10 for anxiety symptoms. | .88 |
| Anxiety | 0–3 | | | .86 |
| PTSD | 0–3 | PTSD Checklist Civilian–PCL–C (Miles et al., 2008) | 16 items assessing symptoms of posttraumatic stress disorder (PTSD) in a civilian population. | .90 |
| Generalized distress | 1–5 | Kessler–6 (Kessler et al., 2003) | Six items assessing generalized psychological distress. | .76 |
| Functional impairment | 1–5 | WHO Disability Assessment Schedule –WHO–DAS–12 (Vázquez–Barquero et al., 2000) | 12 items assessing impaired ability to function across six life domains (household, cognitive, mobility, self–care, social, society). | .83 |
| Community resilience | 1–5 | Escala de Resiliencia Comunitaria (Ruiz Pérez, 2015) | 14 items measuring perceived communal coping and collective self–efficacy. | .81 |
| *Secondary outcome measures* | | | | |
| Well–being | 0–10 | Personal Wellbeing Index (International Wellbeing Group, 2013) | Seven items assessing perceived quality of life (life, health, relationships, security, community connection, future security). | .73 |
| Coping strategies | 1–4 | Brief Cope Questionnaire (Carver, 1997) | Fourteen two–item subscales assessing the use of various coping strategies. | .85 |

Cronbach α: Cronbach alpha for current sample calculated for all participants using PRE assessment data.

cross-study initiative designed to measure the same constructs across studies. Spanish versions of measures were used when available (WHODAS; Escala de Resiliencia Comunitaria); otherwise, measures were translated to Spanish and then back-translated to check translation quality by the research team. The depression, anxiety and PTSD tools (HSCL-25 and PCL-C) had been translated and used previously by the ACOPLE project.

Assessment interviews were conducted by CPAs in Spanish using the KOBO Toolbox platform on tablets. Interviews were conducted in person (with COVID-19 protections in place) or remotely (by phone), in line with modality preferences expressed by the participants. The CPAs had 12 days of training on research interview skills and protocols, including piloting of tools. At post-intervention, 25 in-depth semi-structured individual interviews were conducted with a randomly selected subset of participants, and a focus group discussion was conducted with staff by researchers at the Universidad de Los Andes. The qualitative findings are presented separately (Chaparro Buitrago et al., 2024).

### Statistical analysis

Descriptive statistics were used to assess the balance between trial arms and differences between modalities at baseline (t-test for continuous variables and Chi Squared for categorical variables). Treatment effects were estimated with an intent-to-treat (ITT) and a per-protocol (PP) approach (Thabane et al., 2013). Multilevel regression models with random intercepts were used to represent intraindividual variance across time and to test the effect of intervention, the interaction of assignment to intervention (intervention vs. control) and time (pre vs. post) were included as fixed effects.

For the ITT approach, multilevel models were estimated with data from all participants that were randomized to the control and treatment groups. Missing data in outcome variables was addressed using maximum likelihood estimation (Sullivan et al., 2018). For the PP approach, multilevel models were estimated with data from

participants that completed at least four intervention sessions (based on discussion with the intervention team). Multilevel models were conducted using the package lme4 (Bates et al., 2014) in R statistical software (R Core Team, 2023). The effect size measure presented for the estimated treatment effect is the partial eta squared ($\eta^2_p$) and the effect size presented for the random intercept of individuals is the intra-class correlation.

Sensitivity analyses were performed to explore if changes in estimation resulted in different results. First, PP analyses were rerun while changing the cut-off for inclusion based on number of sessions attended (zero to six) (Supplementary Material S2). Second, inverse probability weighting (IPW) was used to estimate the PP results with a four-session cut-off by adding weights at the individual level. Weighting was done based on predictors of attendance for each modality trial (see Supplementary Material S3 for a detailed statistical analysis). Finally, multilevel models including the community intervention group as a third level variable built to identify the effects of clustering by intervention group on outcomes.

Finally, moderation analyses were performed with the PP in-person and remote samples to explore whether demographic and baseline levels of outcome variables moderate treatment effects. Moderation analyses consisted of linear regression with the change scores (post-scores minus pre-score) of primary outcomes as dependent variables, and the predictors were the interaction between intervention and moderator.

### Ethical considerations

Ethical approval was received from the institutional review boards at both HAI and Universidad de Los Andes. CPAs conducted individual consent processes with each participant, including providing hard copy consent forms that they read aloud to participants. Participants provided verbal informed consent, which was recorded by CPAs. Use of verbal rather than written consent allowed the team to avoid recording participant names on consent forms (the

only place in which names would have been recorded) and therefore better protect confidentiality, which is especially important to victims of the armed conflict who may have concerns about being identifiable or monitored. This approach also simplified the process of gathering consent for remote participants contacted by phone. All participant data were identified using codes and stored on secure research team computers. The post-intervention interview included items selected from the RRPQ to assess participants' reactions to the research process and monitor for potential negative effects.

## Results

### Sociodemographic and baseline measures

Demographic and baseline outcome measures for in-person and remote group participants overall and in the experimental and control groups are presented in Table 1. Participants who chose to participate in-person were significantly older ($M = 40.8$ years, $SD = 16$) than those who chose to participate in remote modality ($M = 35.4$ years, $SD = 11.8$, $p = .001$). Almost all (97%) of the participants in the remote modality lived in urban areas, in contrast to 78% of those in the in-person modality ($p < .001$). There were more unemployed and informally employed participants in the in-person modality and more participants who were formally employed or work at home (domestic duties/childcare) in the remote modality ($p < .03$). At baseline, those in the remote modality reported more functional impairment on the WHODAS than in-person participants ($M_{\text{in-person}} = 1.51$, $M_{\text{remote}} = 1.66$, $p = .04$).

There were no significant differences in sociodemographic variables between control and treatment groups in the in-person or remote trials. Regarding baseline outcome variables, in-person participants reported a higher baseline level of depression ($M = 1.18$, $SD = 0.61$) than the control group ($M = 0.98$, $SD = 0.55$, $p = .03$). In the remote trial, there were no differences between the intervention and control groups for primary outcomes at baseline.

### Feasibility

#### Intervention attendance
In the in-person trial, most participants (62.2%) attended four or more sessions. A quarter (24.4) attended zero sessions, while 13.4% attended one to three sessions. In the remote trial, most participants (65.4%) attended four or more sessions, 11.5% attended zero sessions and 23.1% attended one to three sessions (see Table 3).

#### Predictors of attendance
In the in-person modality, older age positively predicted attendance ($ß = 0.07$, $p < 0.001$, $R^2 = 0.10$). For the remote modality, being employed ($ß = 3.11$, $p = 0.01$, $R^2 = 0.18$) or working to take care of one's household ($ß = 2.08$, $p = 0.03$, $R^2 = 0.18$) positively predicted attendance compared to being unemployed.

#### Fidelity
The mean fidelity score for groups in the in-person modality was 96% (range 91% to 99%) and for the remote modality was 97% (range 89% to 100%).

### Acceptability

#### Intervention acceptability
Participants in both the in-person and remote modalities reported being highly satisfied with the intervention overall. Participants reported that they felt the sessions were private, secure, comfortable, allowed them to feel supported and heard, and were culturally respectful. In-person participants indicated that they felt more comfortable expressing themselves emotionally in sessions than did remote participants (diff = 0.22, $p = 0.04$) and, at trend level, reported more risk to safety than did remote participants (diff = 0.33, $p = 0.05$) (see Table 4).

#### Research acceptability
On the RRPQ-R, participants in both modalities were generally positive about their participation in the research. Those in the in-person trial reported better understanding of the informed consent (diff = 0.12, $p = 0.02$) and were more likely to believe that the research would be useful for others (diff = 0.34, $p < 0.001$) than those in the remote trial (see Table 4).

### Intervention effect estimation

#### In-person trial
Significant in-person ITT intervention effects were found for symptoms of depression with $-0.18$ ($p = 0.03$, $\eta^2_p = 0.03$); for anxiety with a $-0.19$ change ($p = 0.03$, $\eta^2_p = 0.03$) and PTSD with a $-0.27$ change ($p < 0.001$, $\eta^2_p = 0.05$). Consistently, significant in-person PP intervention effects were found for depression with $-0.25$ change ($p = 0.01$, $\eta^2_p = 0.06$); anxiety with $-0.26$ change ($p = 0.01$, $\eta^2_p = 0.06$) and PTSD with $-0.29$ change ($p = 0.01$, $\eta^2_p = 0.06$). Effect sizes for PP were moderate and larger than those in the ITT approach. No effects were found for secondary outcomes in the ITT or PP sample (see Table 5).

#### Remote trial
No intervention effects were found in the ITT or PP analyses for the remote modality (see Table 5).

**Table 3.** Attendance by modality

| | | Number of sessions attended[a] | | | | | | | | | |
|---|---|---|---|---|---|---|---|---|---|---|---|
| | | 0 | 1 | 2 | 3 | 4 | 5 | 6 | 7 | 8 | Four or more sessions |
| In–person | N | 20 | 3 | 2 | 6 | 7 | 11 | 12 | 16 | 5 | 51 |
| | % | 24.4 | 3.7 | 2.4 | 7.3 | 8.5 | 13.4 | 14.6 | 19.5 | 6.1 | 62.2 |
| Remote | N | 6 | 4 | 5 | 3 | 2 | 8 | 12 | 6 | 6 | 34 |
| | % | 11.5 | 7.7 | 9.6 | 5.8 | 3.8 | 15.4 | 23.1 | 11.5 | 11.5 | 65.4 |

[a]This is the total number of sessions attended by participants. For example, a participant that attended sessions 1, 3, 5 and 7 would be classified under "4" (and 4 or more sessions).

**Table 4.** Intervention and research acceptability

| | In-person | | Remote | | In-person vs. remote *t* test *p*-value |
|---|---|---|---|---|---|
| | *n* | Mean (SD) | *n* | Mean (SD) | |
| *Intervention acceptability* | | | | | |
| Accessibility of group | 70 | 2.13 (1.31) | 46 | 2.22 (1.11) | 0.70 |
| Felt comfortable in group | 65 | 3.92 (0.41) | 45 | 3.87 (0.55) | 0.56 |
| Privacy in group | 65 | 3.12 (1.34) | 45 | 3.22 (1.24) | 0.69 |
| Risk to safety in group | 65 | 1.46 (1.09) | 45 | 1.13 (0.63) | 0.05 |
| Felt protected from Covid–19 | 65 | 3.74 (0.69) | 45 | 3.51 (1.1) | 0.22 |
| Felt supported | 65 | 3.92 (0.32) | 45 | 3.89 (0.38) | 0.62 |
| Felt heard | 66 | 3.89 (0.43) | 46 | 3.93 (0.33) | 0.57 |
| Felt needs understood | 65 | 3.86 (0.46) | 46 | 3.8 (0.54) | 0.56 |
| Group was respectful of the culture | 65 | 3.85 (0.51) | 46 | 3.83 (0.53) | 0.84 |
| Comfortable expressing self emotionally | 64 | 3.94 (0.39) | 46 | 3.72 (0.62) | 0.04 |
| Felt confidentiality protected | 65 | 3.97 (0.17) | 46 | 3.89 (0.38) | 0.20 |
| Learned useful skills/tools | 65 | 3.74 (0.62) | 46 | 3.76 (0.57) | 0.84 |
| Overall satisfaction | 65 | 3.95 (0.21) | 46 | 3.96 (0.29) | 0.96 |
| *Research acceptability (RRPQ–R)* | | | | | |
| Would participate again | 157 | 4.17 (0.64) | 86 | 4.15 (0.64) | 0.81 |
| Understood consent form | 156 | 4.25 (0.43) | 89 | 4.13 (0.34) | 0.02 |
| Believe responses will be kept confidential | 156 | 4.31 (0.5) | 88 | 4.2 (0.43) | 0.09 |
| Research is useful for others | 157 | 4.26 (0.56) | 83 | 3.92 (0.86) | 0.001 |
| Experienced intense emotions | 138 | 3.26 (1.12) | 73 | 3.25 (0.98) | 0.92 |
| Procedures were too long | 149 | 2.34 (0.87) | 73 | 2.37 (0.92) | 0.79 |

*Note*: Intervention acceptability was measured using a 4-point Likert scale from 0 = not at all to 3 = a lot. Research acceptability scale used a 4-point Likert scale from 1 = totally disagree to 4 = totally agree.

**Table 5.** ITT and PP treatment effect estimates for primary outcomes in the in-person and remote modality

| | ITT | | | | PP | | | |
|---|---|---|---|---|---|---|---|---|
| Model | Treatment effect | *p*-value | $\eta^2_p$ | ICC | Treatment effect | *p*-value | $\eta^2_p$ | ICC |
| *In–person modality* | | | | | | | | |
| Depression | **−0.18** | **0.03** | **0.03** | 0.62 | **−0.25** | **0.01** | **0.06** | 0.62 |
| Anxiety | **−0.19** | **0.03** | **0.03** | 0.63 | **−0.26** | **0.01** | **0.06** | 0.66 |
| PTSD | **−0.27** | **0.001** | **0.05** | 0.57 | **−0.29** | **0.01** | **0.06** | 0.58 |
| Generalized distress | −0.09 | 0.54 | 0.00 | 0.40 | −0.20 | 0.23 | 0.01 | 0.39 |
| Functional impairment | −0.04 | 0.56 | 0.00 | 0.56 | −0.09 | 0.21 | 0.01 | 0.52 |
| Community resilience | 0.04 | 0.53 | 0.00 | 0.58 | 0.02 | 0.81 | 0.00 | 0.53 |
| *Remote modality* | | | | | | | | |
| Depression | 0.09 | 0.50 | 0.00 | 0.44 | 0.04 | 0.81 | 0.00 | 0.42 |
| Anxiety | 0.03 | 0.81 | 0.00 | 0.49 | −0.04 | 0.79 | 0.00 | 0.50 |
| PTSD | 0.16 | 0.22 | 0.02 | 0.59 | 0.19 | 0.17 | 0.02 | 0.63 |
| Generalized distress | −0.06 | 0.73 | 0.00 | 0.33 | −0.09 | 0.66 | 0.00 | 0.31 |
| Functional impairment | −0.01 | 0.91 | 0.00 | 0.59 | −0.07 | 0.60 | 0.00 | 0.61 |
| Community resilience | 0.22 | 0.15 | 0.02 | 0.19 | 0.29 | 0.11 | 0.03 | 0.17 |

*Note*: Results presented in bold show significant treatment effect. *p*-values for treatment effects are presented in the *p*-value column.

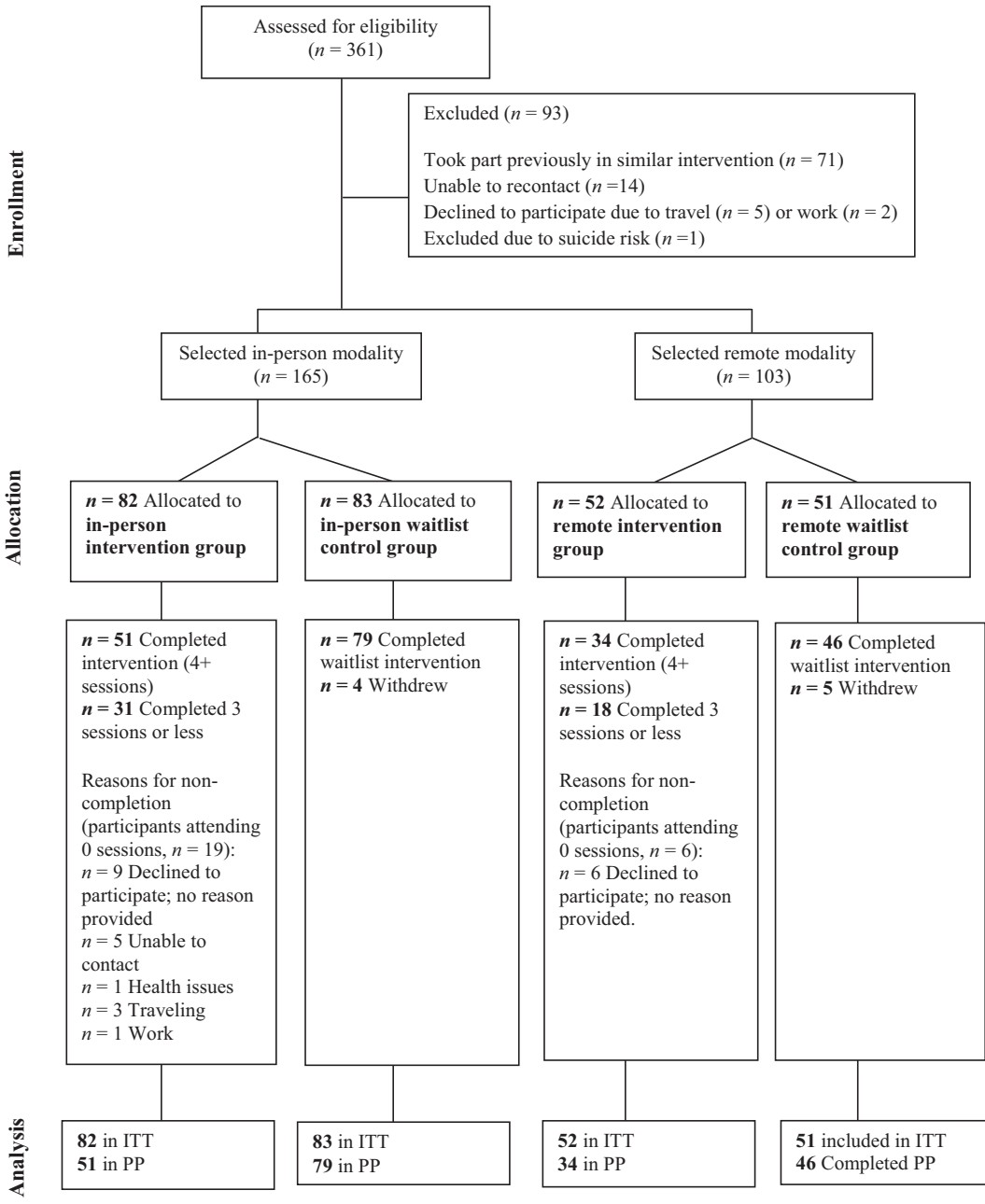

**Figure 1.** CONSORT diagram.

### Comparison of pre- and post-intervention outcomes

Pre- and post-intervention means, SDs and significance levels for *t* tests comparing pre- and post-intervention means are presented in Supplementary Material S1.

### Sensitivity analyses for treatment effects

#### Session attendance sensitivity analysis

Consistent statistically significant results were found for depression, anxiety and PTSD outcomes across all attendance cut-offs in the in-person modality, with small to moderate effect sizes ($\eta^2_{p\ depression} = 0.04$ to $0.10$; $\eta^2_{p\ anxiety} = 0.03$ to $0.06$; $\eta^2_{p\ PTSD} = 0.04$ to $0.07$). There were no consistent significant effects in the

sensitivity analysis for the remote modality (see Supplementary Material S2).

#### Inverse probability weighting

Results for the in-person trial showed significant reduction for PTSD, depression and anxiety but the effect sizes tended to be bigger and closer to a large effect size in comparison to the PP analysis without IPW. Analysis showed no significant results in the remote trial (see Supplementary Material S3).

#### Effects of clustering by intervention group

Multilevel models with individual and group random intercepts were built to explore if clustering by intervention groups explain

variance in the primary and secondary outcomes. There were no significant changes when including group as a random variable.

### Moderation analyses for treatment effects

In-person trial moderation analyses found that living in an urban area was associated with increase in depression post treatment, resulting in no change from pre to post between control and experimental groups (treatment change = $-0.64$, $p < 0.001$, moderation effect = 0.51, $p = 0.02$). Additionally, higher baseline levels of PTSD were associated with increased improvement in anxiety symptoms (treatment change = 0.20, $p = 0.31$, moderation effect = $-0.53$, $p < 0.001$).

In the remote trial, there were no moderation effects for demographic variables. Higher baseline levels of depression were associated with an increase in PTSD symptoms resulting in no change for PTSD in the treatment group (treatment change = $-0.62$, $p = 0.04$, moderation effect = 0.62, $p = 0.01$).

### Discussion

This study assesses feasibility, acceptability and preliminary effectiveness outcomes of a CB-PSS group intervention for conflict-affected adults in Colombia's Pacific Coast during the COVID-19 pandemic and during the national strikes in Colombia. In doing so, it aims to contribute to the existing evidence base and practitioner guidance regarding community-based interventions conducted by nonprofessional community members, using varied (in-person and remote) modalities to accommodate contextual challenges. In-person and remote versions of the same intervention were assessed in parallel trials, with participants able to choose which modality to participate in. Results suggested overall feasibility and high acceptability in both trials and preliminary evidence of effectiveness in the in-person trial. Findings are discussed in detail below in the context of companion papers documenting qualitative findings on barriers and facilitators from the same study (Chaparro Buitrago et al., 2024) and results of a previous pilot study (Rattner et al., 2023).

Because participants were able to decide whether to join the in-person or remote trials, initial analyses were conducted to identify demographic and baseline mental health characteristics of those selecting each modality. Participants who chose to participate in-person were older and more likely to live in rural areas than remote participants, who were almost entirely based in urban areas. In-person participants were also more likely to be unemployed or informally employed, while remote participants were more likely to be formally employed or work at home to take care of households and children. It may be that younger, urban participants were more likely to be tech-savvy and therefore to prefer a remote group modality, a finding that aligns with research identifying remote service implementation challenges related to older age, technological literacy and poor connectivity (Ibragimov et al., 2022; Witteveen et al., 2022). Poor internet access in rural areas may also have discouraged remote participation. Those with formal employment, or those taking care of children and the household may also have appreciated the flexibility of remote participation. Remote participants also reported more functional impairment related to mental health reactions than in-person participants, but baseline rates of functional impairment were very low in all groups. These findings may be useful for practitioners in determining when and for whom remote or in-person interventions are preferable.

In this study, *feasibility* was assessed by means of participant attendance levels and intervention fidelity. *Attendance* was moderate for all participants, with 62–65% attending four or more sessions in both modalities. Fewer participants attended zero sessions in the remote modality than in the in-person modality. It is possible that those who did not attend in-person sessions were more likely to be impeded by travel and logistical constraints which prevented any attendance at all. Among in-person participants, older participants had better attendance; these individuals may be more motivated to attend this type of intervention or have more means to do so and fewer competing priorities. In the remote modality, those who were employed or took care of their households were more likely to attend than unemployed participants, who may have more unpredictable schedules due to efforts to secure income, especially during the pandemic and strikes when livelihoods were particularly threatened. Results are consistent with qualitative interviews (Chaparro Buitrago et al., 2024) in which participants and staff described challenges in in-person groups associated with travel difficulties, COVID-19 infection, last minute work commitments and childcare (especially for women), while remote participants struggled with unstable internet, power outages and poor technological literacy. In sum, both modalities appeared moderately feasible even in the midst of multiple emergencies, with neither demonstrating significantly greater advantages regarding attendance than the other. Participants were able to select their modality, and it is possible that were they not able to do so, attendance may have suffered further –an empirical question that should be explored further in future studies.

Intervention *fidelity*, assessed through fidelity checklists completed by psychologist and social worker observers, was similarly high in both modalities, suggesting that, even in the remote modality, group facilitators were generally able to complete key intervention activities. Although results are encouraging, fidelity checklists did not assess quality of facilitation beyond whether or not activities were completed or level of participant engagement with activities. Future research should more comprehensively examine intervention fidelity across modalities.

Participant reported *acceptability* was high in both in-person and remote trials, a finding consistent with the pilot study results and qualitative results from the current study. In all three datasets, both in-person and remote participants reported strong satisfaction with the intervention. However, in qualitative interviews, participants described a collaborative, peer support dynamic in both modalities (Chaparro Buitrago et al., 2024), while in the pilot study, participants highlighted a more collaborative style in the in-person groups (Rattner et al., 2023). Data from the current research supports both findings, with both in-person and remote participants reporting similarly high levels of feeling comfortable, supported, heard and understood, but in-person participants indicating that they felt more comfortable expressing themselves emotionally in sessions than did remote participants. In-person participants also reported better understanding of the consent form and greater belief that research results would be helpful to others, suggesting better overall engagement and perceived benefit. These findings may have implications for potential effectiveness, discussed further below.

Preliminary *effectiveness* outcome analyses revealed significant reduction in symptoms of anxiety, depression and PTSD for in-person participants in comparison to control group participants. Results were consistent in both the ITT and PP analyses, though effect sizes were higher in the PP sample comprised of participants attending four or more sessions of the eight-session intervention.

These findings are consistent with other research showing positive results of task-sharing and CB-PSS interventions conducted in-person (Raviola et al., 2019; Le et al., 2022), including previous study of ACOPLE's group and individual models implemented by lay community workers in Colombia (Pacichana-Quinayáz et al., 2016; Osorio-Cuellar et al., 2017).

However, in the in-person trial, no significant differences between the intervention and control group participants were found for other measures, including generalized distress, functional impairment, community resilience or for secondary outcomes. Of note, most of these measures were added as part of a cross-study initiative to measure parallel constructs across interventions and contexts, and had not been used previously with this project, whereas anxiety, depression and PTSD measures had been used previously with the ACOPLE project for monitoring and evaluation purposes; therefore, CPAs may have been more accustomed to their use and explaining these items to participants. Examination of the pre- and post-intervention means can further aid interpretation (see Supplementary Material S1). Regarding generalized distress, in both trials, both intervention and control participants demonstrated improvement over the course of the study period, but there was no difference between the groups at post-intervention. By the end of the study, circumstances had improved regarding both COVID-19 and nationwide protests. It is possible that these environmental changes resulted in reduced generalized distress for all, such that intervention effects were only evident in the more extensive clinical measures among participants experiencing distress beyond the 'typical' level. Indeed, moderation analyses suggest that the in-person intervention was more effective for those with significant distress; those with higher levels of PTSD at baseline were more likely to benefit from the intervention in regard to anxiety. In the case of functional impairment, in both in-person and remote trials, pre- and post-intervention means suggest a potential floor effect. Rates of impairment were very low at baseline and remained low throughout, implying that this sample may not have experienced sufficient levels of impairment to fully respond to this intervention.

Participants in the remote trial showed no change in mental health outcomes in comparison to control participants. Examination of pre- and post-intervention means reveal that in the remote trial, participants in both the intervention and control conditions showed significant improvement in depression, anxiety and PTSD symptoms over the course of the study, resulting in no difference between conditions at post-test (Supplementary Material S1). As mentioned, the negative effects of COVID-19 and the protests had lessened by the end of the study. It is possible that these environmental changes resulted in reduced generalized distress for all (as described above) and a reduction in anxiety, depression and PTSD for remote participants, washing out potential intervention effects. Although it is not clear why the mental health of control participants in the remote trial would improve more than that of in-person control participants, it possible that characteristics of those who chose to participate remotely may play a role. For example, remote participants were likely to be younger and reside in urban areas – a demographic perhaps more affected by COVID-19 and national protests, who may therefore have benefited more from an improvement in related conditions. Indeed, moderation analyses suggest that the in-person intervention did not change depression levels among those in urban areas, perhaps because they benefited from simultaneous improvement in contextual factors during the study period. These and other potential implications of non-

randomization into in-person and remote modalities are discussed further in the Limitations sections.

Although methodological considerations mean that results should be viewed with caution, potential explanations for differences in outcomes between in-person and remote groups can be gleaned from qualitative data shared in accompanying articles (Rattner et al., 2023; Chaparro Buitrago et al., 2024). As mentioned earlier, remote groups faced challenges regarding frequent loss of mobile signal and electricity, especially during heavy rains, which, according to participants and staff, reduced connetivity thus impeding engagement during the sessions. Second, remote participants and CPAs reported concern about distractions and compromised confidentiality due to participants attending sessions while doing other activities (e.g., working, studying housework), sometimes in the vicinity of family members or work colleagues. This occurred despite efforts by CPAs to encourage attendance in quiet and private settings as part of the remote services safety planning process prior to the intervention. These factors had potential to impede engagement and development of a safe and confidential group dynamic. Finally, in the pilot study, some staff and participants mentioned that certain activities were difficult to adapt to the remote format (e.g., dance and other expressive movements) and that some forms of peer support may also have been harder to enact in the remote setting. Indeed, remote participants in the current study reported less comfort expressing their emotions in the groups than in-person participants, suggesting that such factors may have impeded emotional expression and therefore potentially dampened intervention benefits. In-person participants also reported better understanding of the consent form and greater belief that research results would be helpful to others, suggesting that they found the research process more palatable than remote participants, which may have influenced engagement with these elements. Similar processes have been suggested by other studies (Naslund et al., 2019; Ibragimov et al., 2022) and suggest that future intervention work may benefit by exploring methods of strengthening remote interventions, including by facilitating engagement, emotional expression, exchange of peer support and confidentiality in remote settings.

## Limitations

Methodological issues should be considered when interpretating these results. First, the sample size for remote groups was relatively small, raising concern about insufficient power to detect results. However, the data do not support this interpretation. Rather, in the remote modality, both intervention and control participants demonstrated significant within-group improvement in symptoms during this study period and the intervention group did not improve more than the control group. Additionally, this study was implemented with a sample made up of mostly women and group sessions were facilitated by women CPAs. With such a limited sample of men, this study cannot reliably speak to feasibility, acceptability or effectiveness among men, or to gender differences.

More significantly, although this study includes exploration of differences between in-person and remote trials, it is critical to bear in mind that participants were not randomized to in-person or remote modalities, but rather were given the option to choose which modality they preferred (then randomized to experimental and control conditions). This decision was made with the aim of increasing equitable access while also prioritizing participant autonomy and decision-making and in response to participant feedback during the pilot study expressing that being given options

regarding modality was highly valued during the COVID-19 emergency. However, this approach introduces the possibility of selection bias, such that characteristics of participants choosing a certain modality may be characteristics that also make them more likely to attend (or to drop out) and to benefit (or not) from the intervention. To partially explore this possibility, moderation analyses were conducted to determine whether demographic or baseline mental health factors influenced outcomes. In-person and remote participants showed significant differences in age, work status and residing in urban versus rural areas. Moderation analyses revealed that in the in-person trial, those living in urban areas showed no change in depression (while participants in rural areas did). As most remote participants reside in urban areas, this factor may contribute to explaining the results. Other individual-level factors not measured here, such as participant motivation and value attributed to peer support, problem-solving and emotional regulation, may also play a role, as well as exposure to confounding variables such improvement in COVID-19 and national strike conditions (as discussed earlier in this section).

These studies were implemented during highly challenging circumstances, at the height of the COVID-19 pandemic and in a period of national strikes and police violence. These elements affected attendance and resulted in delays in intervention implementation which altered methodology, likely impeded engagement and momentum, and may have impacted outcomes. Pauses and delays due to COVID-19 infections among staff and participants (approximately three weeks in most cases) affected in-person groups more than remote groups, but remote groups were also delayed when facilitators became ill.

Overall, considering partial results, methodological limitations and contextual challenges, results, particularly those regarding effectiveness, should be viewed with caution, and additional research is needed to attempt to replicate findings and further explore explanations. Future research conducted in nonemergency contexts should include randomization to in-person and remote modalities.

## Conclusions

In this study, a CB-PSS group utilizing task sharing to nonspecialist community members was found to be moderately feasible and highly acceptable when implemented in-person and remotely. The intervention was found to be effective in reducing some forms of distress when implemented in-person; however, no effects were found when the intervention was conducted in a remote modality using calls through an online platform. These preliminary results suggest that trained community workers can meaningfully improve the mental health of their peers when engaging in person but that, while remote approaches can be an important strategy for facilitating engagement, additional research is needed to assess effectiveness. Future research should be conducted in more stable conditions and include randomization of participants to in-person and remote modalities to replicate and further interpret the results.

**Open peer review.** To view the open peer review materials for this article, please visit http://doi.org/10.1017/gmh.2024.50.

**Supplementary material.** The supplementary material for this article can be found at http://doi.org/10.1017/gmh.2024.50.

**Data availability statement.** Data are available from authors upon reasonable request.

**Acknowledgments.** The authors would like the acknowledge the research participants and the CPAs (Mariluz Andrade Becerra, Yenis Orejuela Cuesta, Yaila Mena del Pino, Cindy Martinez Duran, Elizabeth Mena, Sandra Murray, Ingrid Murillo Palacios and Ana Perez Perea) from Quibdó, Colombia, all of whom went to great lengths to facilitate this study amidst COVID-19 and national protests and demonstrated impressive resilience and resourcefulness throughout. The authors also acknowledge the contribution of Heartland Alliance International's mental health professionals, Danny Mena and Ana Odilia Lizcano, who provided critical clinical oversight for this study. Finally, we thank Esteban Moreno, Colombia Country Director and Ibeth Lorena Salazar, Monitoring and Evaluation Officer, at Heartland Alliance International, for their contributions to make this research possible.

**Author contribution.** L.E.J. is the principal investigator and led conceptual development and manuscript development. N.G.M. led data analysis, interpretation and supported manuscript development. J.F.B.G supported study implementation and oversight of data collection. M.R. supported study implementation and manuscript development. All authors reviewed and approved the final manuscript prior to submission.

**Financial support.** This study was funded by the United States Agency for International Development (USAID) under the Health Evaluation and Applied Research Development (HEARD), Cooperative Agreement No. AID-OAA-A-17-00002. This study is made possible by the support of the American People through USAID. The findings of this study are the sole responsibility of the authors and do not necessarily reflect the views of USAID or the United States Government.

**Competing interest.** The authors declare none

**Ethics statement.** Study procedures were approved by Institutional Review Boards (IRBs) at Universidad de los Andes and Heartland Alliance International. All participants provided informed consent.

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
