## [Editor Report]

Thank you for the opportunity to review “Effects of a community-based group psychosocial support model for conflict survivors in Colombia: A test of in-person and remote intervention modalities during the Covid-19 pandemic”. I found the article interesting and appropriate for the audience of Global Mental Health. However, several issues should be addressed, including major concerns about the study design and methodology, with serious implications for the results presented. Given this significant limitation, I think this study needs to be reframed as a pilot study, focusing on acceptability and feasibility of the intervention in each of its modalities, and presenting preliminary efficacy or each, separately.

Other comments by section are as follows.

Abstract:

Well written and clearly conveys a summary of the study

Introduction:

- Literature review situates the study within current research. Question is well articulated.

- Line 27: please introduce MHPSS when first used

Methods:

- Please mention the intervention type/name in the study design section

- More details are needed about the sampling method. How did the chain-referral work? How was the sample size decided?

- Related, given the snowball sampling approach, more details on the population are needed. Where were participants engaged? (e.g.at home, community center, etc.), were participants already engaged with other health services with CPAs?

- What is a CPAs and what type or role/programs do they play in the community?

- How was exposure to conflict ascertained?

- How was suicidality/self-harm/psychosis assessed?

- I am puzzled as to why participants were given the option to choose the intervention modality? Why wasn’t that randomized as well? This seriously affects the rigor of the study design as it detracts from a randomized controlled trial. Randomizing only to immediate vs. delayed intervention delivery doesn’t eliminate biases that randomization is intended to do.

-Why was randomization stratified by neighborhood? How many neighborhoods were there?

-The intervention and its development needs to be more fully described. The references provided only describe the qualitative study after the intervention was delivered in the present study (Chaparro et al 2023) or are related to the adaptation of ACOPLE but have not been published and are unavailable (Rattner et al 2023).

-PM+ consists of 5 sessions, what amount of content from these informed the 3 sessions in this study? Why only 3 sessions? Who was in charge of ACOPLE’s ‘evolution’ to a community problem solving intervention?

-How and why where the measures and outcomes selected? Were these translated? Adapted? If so, by who?

-Why were participants only assessed for risk behaviors at the PRE assessment and not in the POST?

-How were moderators selected and tested?

-What were the psychosocial support groups and why was clustering evaluated?

Results:

-Table 1 should not include p-values. The p-value is a measure for inferential purposes, not for descriptive ones. Any differences encountered in baseline characteristics are the production of randomization and hence are not informative.

-Table 2 requires a full heading describing contents. Table is not cited in the text

-Why are both intervention modalities presented together as ‘general sample’? I don’t think you can treat these as the same intervention without evidence of equivalent efficacy.

-Intervention attendance: again, I don’t understand why both intervention modalities were evaluated together. It would seem obvious to me that each modality would have difference attendance and that this information would be of value? Why wasn’t this evaluated?

Discussion:

-Second paragraph states that ‘participants who attended four or more sessions demonstrated significant improvements”. Which intervention modality is this referring to?

-How does the intervention presented here compare to other community-based interventions that have been tested in Colombia? (Aranguren42 Romero & Rubio-Castro, 2018; Osorio-Cuellar et al., 2017; Pacichana-Quinayáz et al., 2016).

-The reasons why intervention modality was given as a choice to participants are not substantiated. The adapted intervention presented here and each of the modalities had been tested before, hence, assuming that modalities were comparable is a far reach.

-Although results by intervention modality are discussed and methodological differences stated, these are not sufficiently addressed. Authors fail to recognize the possibility of unmeasured biases beyond demographic characteristics arising from a non-randomized approach.